# Multivariate Analysis of Risk Factors of the COVID-19 Pandemic in the Community of Madrid, Spain

**DOI:** 10.3390/ijerph18179227

**Published:** 2021-09-01

**Authors:** Víctor Pérez-Segura, Raquel Caro-Carretero, Antonio Rua

**Affiliations:** 1University Institute of Studies on Migrations, Comillas Pontifical University, 28015 Madrid, Spain; 2Industrial Organization Department, ICAI-School of Engineering, Comillas Pontifical University, 28015 Madrid, Spain; rcaro@icade.comillas.edu; 3Department of Quantitative Methods, Comillas Pontifical University, 28015 Madrid, Spain; rvieites@icade.comillas.edu

**Keywords:** COVID-19, the community of Madrid, environmental and socioeconomic risk factors, principal component analysis, cluster analysis, general linear model

## Abstract

It has been more than one year since Chinese authorities identified a deadly new strain of coronavirus, SARS-CoV-2. Since then, the scientific work regarding the transmission risk factors of COVID-19 has been intense. The relationship between COVID-19 and environmental conditions is becoming an increasingly popular research topic. Based on the findings of the early research, we focused on the community of Madrid, Spain, which is one of the world’s most significant pandemic hotspots. We employed different multivariate statistical analyses, including principal component analysis, analysis of variance, clustering, and linear regression models. Principal component analysis was employed in order to reduce the number of risk factors down to three new components that explained 71% of the original variance. Cluster analysis was used to delimit the territory of Madrid according to these new risk components. An ANOVA test revealed different incidence rates between the territories delimited by the previously identified components. Finally, a set of linear models was applied to demonstrate how environmental factors present a greater influence on COVID-19 infections than socioeconomic dimensions. This type of local research provides valuable information that could help societies become more resilient in the face of future pandemics.

## 1. Introduction

The first cases of coronavirus were identified in late 2019 in Wuhan Province, China. On 7 January 2020, the virus was isolated for the first time, verifying the existence of a new strain of coronavirus which became known as SARS-CoV-2 [1]. In a short time, the pandemic spread to six continents, achieving global significance. However, the impact of the pandemic has been uneven between countries. While some nations have hardly been affected, others, such as Spain and Italy, have suffered disproportionately heavy impacts. This difference is partly explained by the different non-pharmaceutical intervention strategies implemented by different nations [2]. However, there are still many unknowns in explaining these significant differences between countries. These gaps in our understanding have propelled increased scientific research seeking to answer a key question: what risk factors make societies vulnerable to the COVID-19 pandemic?

A prolific line of research has been related to the evaluation of environmental factors (such as weather conditions or pollution levels) as mediators of contagion [3,4]. Despite the large amount of research on this subject, it is far from being fully understood. As noted by the metatheoretical studies of Shakil et al. [3] and Briz-Redón and Serrano-Aroca [4], this body of work lacks uniformity in its results. This lack of consistency stems from several different factors, such as the differences in how variables are measured, the analytical strategies employed, the factors under consideration, and the spatiotemporal frame studied. Both studies also point to another common methodological limitation: the overuse of bivariate analyses (i.e., Pearson and Spearman correlations) and descriptive techniques to explain the results. Given the difficulty of reaching consistent and universal conclusions and recognizing the urgency of understanding the environmental factors involved in COVID-19 propagation, this study aims to overcome the limitations of the existing literature. The study achieves this through three core methodological considerations: (1) a complex causal process, (2) a limited space-time frame, and (3) a multivariate methodology.

The complex causal approach comes from taking a broad perspective on environmental factors. It considers socioeconomic conditions, climatological environmental factors, and the level of atmospheric pollution to be parts of the human-environmental ecosystem. This perspective, coupled with the application of multivariate techniques, allows for more refined analytical tools to better understand the pandemic’s behavior. Finally, there is the question of territorial delimitation, addressed through data harmonization and the relevance of specific locations. The resulting work is a case study of the community of Madrid during the first wave of the pandemic. 

This type of micro-level, holistic approach allows us to gain valuable knowledge about local ecological vulnerabilities. Rather than the universalist goal of identifying a general recipe, we opt for an approach that allow us to extract solid and practical information that can be used to make societies more resilient in the face of future disasters.

## 2. Materials and Methods

### 2.1. Methods and Procedures

The main research objective is to empirically ascertain the influence of environmental factors in the case of the Madrid community. Based on this premise, a multivariate approach was chosen. The study methodology consists of different phases. The first phase comprised the identification and selection of risk factors through a literature review. Once the risk factors were identified, a maximum likelihood principal component analysis (PCA) with varimax rotation was applied—which explained 71% of the variance in the original variables—to reduce the set of variables to a smaller number of risk dimensions. The number of components that were retained was determined by performing a parallel analysis, which was a more sophisticated method than most of the classic analysis options [5]. With these new dimensions, a k-means cluster analysis was performed in order to classify the territory according to the new risk variables. To select the most appropriate number of clusters, the elbow method was used in a scree plot of the within-cluster sum of the square of each number of clusters [6] (see Figure A1 in Appendix A). Subsequently, an analysis of variance (ANOVA) was performed and the post hoc Scheffé’s method was used to characterize each cluster according to the original variables. The efficiency of the constructed vulnerability map was verified by another ANOVA, and Scheffé´s multiple comparisons test, according to the number of infections in each territory. The analysis ended with a set of linear regression models used to determine the influence of each component on the total cumulative cases for the whole territory and for each cluster. For some linear models—given the problems of autocorrelation and heteroscedasticity that geographical information usually presents—their regression coefficients were estimated using a covariance matrix consisting of heteroscedasticity (HC) (specifically, the HC3 matrix, which performs better with smaller samples (<250)) [7]. The heteroscedasticity was checked by the White test [8].

### 2.2. Data

The total number of municipalities in the community of Madrid were used as the statistical units for the study, with the exception of the municipality of Madrid itself, which was replaced by the 21 districts that comprise it. This choice was driven by the understanding that the districts of the municipality are more readily comparable with the rest of the municipalities than the entire municipality would be. A total of 199 districts and municipalities were studied. The data were obtained from generally accessible public databases. Furthermore, the study can be replicated and extended in the event of future pandemic outbreaks, since more data on the progress of the pandemic are becoming available as Europe struggles with a third wave of lockdowns. The analysis was restricted to first-wave data from 14 March 2020 (when a lockdown was imposed in Spain) to 13 May 2020 (when lockdown restrictions were eased). 

Firstly, the data on the target variable, the total accumulated infections as of May 13 in the Madrid community, were retrieved from the open data set “COVID 19 -TIA” by the Municipalities and Districts of Madrid [9]. We opted to use the absolute values instead of the ratios per 100,000 because of the large population differences between territories. These differences cause the relative measures to bias the incidence indicator by inflating the influence of smaller regions with hardly any cases, which may be due to small, random outbreaks. This issue is highlighted by the fact that three of the six regions with the highest rates have fewer than seven cases to date (Valdaracete with six cases, and Berzosa del Lozoya and Braojos with less than five cases). In the regression model, population size is used as a control variable for the population size bias related to the absolute measure. 

This variable was transformed prior to the analysis. The first transformation was conducted due to the high incidence of missing data (~21%). In this case, the missing data were due to confidentiality issues, because the information for municipalities with less than six cases was suppressed. The imputation procedure selected is imputation by a constant (=5) according to a parsimony criterion. The second treatment was a Box–Cox logarithmic transformation, used to achieve a higher degree of normality in the distribution.

The set of independent variables are summarized in Table 1. In addition, since the dependent variable is the total number of cases, the population of each territory has been included as a control variable for the linear models. 

### 2.3. Software

The statistical analyses were carried out using the R V4.0.1 language [21] (R Foundation for Statistical Computing, Vienna, Austria) and several libraries, such as “sf” [22], “tidyverse” [23], “psych” [24], “extrafont” [25], “cluster” [26], “sandwich” [27], “lmtest” [28], “reshape2” [29], “factoextra” [30], “agricolae” [31], “ggspatial” [32], and nortest [33]. The open source code and data required to replicate all analyses in this article are available at Pérez-Segura et al. [34].

## 3. Theory Framework

In mid-February, Italy overtook China to become the country with the most COVID-19 cases per million, but was soon overtaken by Spain (25 March 2020), which ranked first in the world until 10 April 2020 [35]. The Spanish capital, Madrid, was the epicenter of the outbreak during the first wave, accounting for most of the infections in the country. Following Tobias’ findings [36], strict confinement measures were the key to reducing the infection rate in Spain to its minimum at the end of April. However, with the arrival of the summer months and the relaxation of mobility restrictions, the downward infection trend reversed. This increase in the number of cases was accentuated by strong outbreaks in Spanish communities that were barely affected previously. This resulted in a second wave that was disproportionately larger than the first, although the mortality rate dropped significantly. Subsequently, the second wave gave way to the third wave, which once again surpassed the previous waves in terms of the number of cases [35], demonstrating the difficulty of controlling the contagion without resorting to intense mobility restriction measures. 

The interest in studying the Madrid community stems from three factors: (1) Madrid is one of the world’s main COVID-19 hotspots, (2) the lack of previous studies focused on the territory, and (3) a geographic space with harmonized information for the whole territory. Moreover, focusing on the first wave provides two other advantages: (1) it was a critical moment during the pandemic, since the degree of initial propagation has an impact on subsequent waves; and (2) it allows easier control of exogenous factors such as partial lockdowns and non-pharmaceutical measures, which were employed in Madrid during the following waves. 

During the first year of the pandemic, much scientific literature was published exploring the influence of environmental factors on levels of infection. The relationship between these variables was previously confirmed with other infectious diseases, such as influenza [37]. Despite the extensive work carried out, the different studies did not produce consistent findings on how certain variables relate to COVID-19. There are many factors that contribute to the lack of coherence between the results, and they can be classified into three categories: the variables considered, divergences in measurement methods, and the applied methodologies.

There are differences in the measurement methods employed for certain factors relevant to the study, which can complicate efforts to make statistical comparisons between them. These factors can be diverse and include areas, such as differences in COVID-19 testing strategies between countries (e.g., the type of test and its administration strategy), the location of the meteorological stations, and the time period studied. In light of this, limiting our study to a specific geographical area is an appropriate strategy through which to avoid this possible source of error.

The number of risk factors considered and the precision with which they are measured is another key element that greatly impacts the study results. The totality of factors influencing the propagation of COVID-19 is still unknown. For example, it was only recently discovered that the virus has a predilection for certain blood types [38]. Similarly, Allcott et al. [39] recently discovered how individual political orientations mediate attitudes towards the use of prevention measures. Davies et al. [40] recently demonstrated how the British COVID-19 variant has increased transmissibility. These studies are examples of the large number of possible latent variables influencing the pandemic, whose impact may be responsible for the lack of consistent results between different studies. Therefore, it is informative to make approximations that cover a wide spectrum of variables, even with the knowledge that there will be many other variables with significant influences that cannot be included. This issue is linked to Briz-Redon and Aroca [4] and Shakil’s [3] considerations about the lack of diversity among methodologies, where bivariate approaches abound, as opposed to multivariate techniques. This idea led to the development of multivariate analytical approaches that allow for the examination of the phenomenon from a broader perspective than monocausal approaches do. Sarkoide and Owusu [41] have recently published research comparing countries on a global scale and employing a multidimensional approach. This work is complementary to ours, as they have a different spatial and temporal framework, as well as a different analytical design. A strength of our present work is that its greater granularity makes it possible to appreciate the heterogeneity within territories. This allows us to extract more reliable knowledge of the concrete reality of each area, in addition to solving the heterogeneity problems.

According to the World Health Organization [42], a risk factor is defined as a characteristic, condition, or behavior that increases the likelihood of suffering a disease or injury. Even though these factors are most often presented individually, they often relate to each other. According to the literature review, risk factors are classified into three dimensions: socioeconomic, pollution, and climatological.

### 3.1. Socioeconomic Dimension

The socioeconomic dimension includes four variables, two related to demographic factors and two related to economic indicators. The demographic factors considered are the age composition of the population (percentage of people over 65 years of age) and the population density. Kang and Jung [43] have pointed out the relationship between the COVID-19 pandemic and certain social groups characterized by age, such as high mortality among the elderly. 

The way in which COVID-19 is transmitted, through close contact between individuals, makes it essential to consider the population density of a territory as a factor influencing the rate of transmission. Therefore, the higher the population density, the easier it is to transmit the disease. This relationship is empirically confirmed in Carozzi’s research [44]. During the first wave, Madrid spent most of the time in a general lockdown. The study by Sun et al. [45] found that, under lockdown conditions, population density does not seem to have a significant effect on infection. However, population density was included in this study because the lockdown was initiated only after infections began to skyrocket.

The economic situation of individuals is another element that has been related to the spread of previous pandemics, such as the Spanish flu and AIDS [46,47]. A certain degree of correlation has also been found with this pandemic. Hawkins et al. [48] noted statistical associations between geographic incidence and the socioeconomic status of communities in the United States when using the Distressed Communities Index as an indicator of economic status (DCI score is composed of items such as the number of adults without a high school degree, the unemployment rate, the poverty rate, etc.).

A lack of economic resources translates into the reduced availability of hygienic/sanitary resources. Additionally, it is also related to poorer living conditions, such as the rate of overcrowding in housing, which is a key factor in maintaining social distancing measures to avoid infection during lockdowns. People struggling with unemployment or limited employment opportunities may be under greater pressure to perform risky actions to keep their jobs or seek economic resources. Ahmed et al. [49] suggests in his theorical work that impoverished populations may have a poorer health status associated with poor living conditions, which increases their vulnerability to COVID-19. The economic factors included in this study are the per capita income of the territory and the percentage of workers supported by social security. The percentage of workers has been included since the average income variable per municipality may be insufficient to reflect the quality of the labor market in a territory. The average does not reflect the possible economic inequality within the territory. Additionally, there may be well-paid, low-skilled jobs that are performed under worse working conditions, implying a lower labor status. Thus, the combination of the two variables makes the socioeconomic characterization of the area more robust.

### 3.2. Pollution Dimension

Different studies agree on the association between air pollution levels and COVID-19, both in relation to infection and mortality [50,51]. Although none of these studies account for the causal mechanism that produces the effect, the concurrence of the results gives us confidence that such a relationship exists.

The pollution dimension is formed by two variables, each of which is based on different ways of evaluating pollution in the different territories. The first is composed of all of the pollutant measurements available in the Madrid community (CO, SO_2_, NO, NO_2_, Ozone, and PM2.5). The second is the variable percentage of deaths associated with respiratory problems, which serves as an indirect measurement of the chronic pollution in the territory. The findings of Zheng et al. [52] verify how long-term exposure to pollutants is related to the number of COVID-19 cases.

### 3.3. Climatological Dimension 

Temperature and humidity are fundamental environmental elements that have considerable influence on infectious respiratory diseases, such as influenza. There is no unanimity on the type of influence they have on COVID-19. Bashir et al. [53] found a positive correlation between temperature and transmission, yet, on the contrary, Ma et al. [54] found a negative correlation between the two. There are even studies, such as that of Mollalo et al. [55], in which there is no association between the variables. Taking into account the results of the bibliographic review of the work of Briz-Redon and Serrano-Aroca [4], the studies that identify a negative correlation (33 papers) outnumber those that find a positive correlation (6 papers) and the publications that find no relationship (7 papers). 

The same inconsistent results are found in relation to humidity, but, considering the work of Briz-Redon and Serrano-Aroca [4], it seems that most of the research where this variable is included tends to present a negative relationship, as opposed to a positive relationship or a lack of any association between the factors (thirteen, three, and six papers, respectively).

## 4. Results and Discussion

Before carrying out the PCA, tests were performed on the data matrix in order to check its suitability for the technique. Bartlett’s test of sphericity showed a *p*-value of 1, so the null hypothesis of the independence of variance was accepted. The Kaiser–Meyer–Olkin measure was also examined, and the adequacy of the data was verified (an overall measure of sampling adequacy of 0.73).

The principal components technique with varimax rotation was applied and three components emerged from the 13 initial variables, which explain 71% of the original variance. Each component was interpreted in terms of factor loadings (Table 2). It should be noted that the empirical dimensions resulting from the analysis do not correspond to the theoretical dimensions projected. This is explained by the fact that previously validated scales were not used. However, the resulting dimensions are still useful insofar as they summarize information from the 13 risk factors considered.

One aspect of the new variables is that the population density variable is loaded onto a mainly climatic factor. This finding makes sense in light of the fact that pollution is an ecological footprint of human populations.

Once the new variables or dimensions were chosen, a k-mean cluster was carried out in order to identify the different ecosystems that comprise the community of Madrid, according to the risk factors (Figure 1). The “Madrid-City” cluster is made up of the 21 districts of the city, and has notably higher scores on measures of pollution, population density, and socioeconomic conditions than the other two (Figure 2). However, in regard to the particulate matter and temperature factors, Madrid has a negative score close to that of the “North-East” cluster. The “Madrid-Surroundings” cluster is composed of a total of 96 municipalities, among which are several heavily populated municipalities, such as Fuenlabrada and Leganes (~200,000 inhabitants) as well as Alcorcón, Parla, and Torrejón de Ardoz (~130,000 inhabitants). In this cluster, the particulate matter and temperature factors stand out. The pollution and population density variables are considerably lower than in the “Madrid-City” cluster, but slightly higher than in the “North-East” cluster. This is also the region with the worst socioeconomic status of the three. Finally, the “North-East” cluster is composed of 82 municipalities, all of which are municipalities with less than 5000 inhabitants. This corresponds to lower pollution levels as well as a lower socioeconomic status.

Table 3 provides more detail on the characteristics of each cluster according to the mean scores of each of the original variables. The ANOVA *p*-values (Table 3) show that there are statistically significant differences between means in at least one cluster for all variables. Consequently, the post hoc test was employed to determine which clusters the differences existed between. The Madrid-City cluster was the most polluted region of the territory, followed by the Madrid-Surroundings cluster, while the North-East cluster stood out for its low pollution levels. The socioeconomic component demonstrates a similar trend, with Madrid-City scoring the highest. Measures of particulate matter and temperature, on the other hand, displayed a completely different trend, with the North-East cluster being the most humid and the Madrid-Surroundings cluster being the hottest. For both variables, Madrid-City had the lowest scores.

After defining the different ecosystems according to the risk factors, we evaluated whether they corresponded to different incidence rates of infection. Figure 3 shows the level of disease penetration per cluster. This graph shows that the municipality of “Madrid-City” suffered the highest incidence of infection. The “Madrid-Surroundings” cluster had a large number of outliers (Figure 3). These outliers correspond to the highly populated cities mentioned above (Leganes, 2963 cases; Alcala de Henares, 2331 cases; Mostoles, 2079 cases, etc.). The ANOVA test and the Scheffé´s method post hoc test showed significant differences between the average levels of contagion in the three territories. These results showed how the use of risk factors could delineate differentially affected areas. However, it did not shed light on the influence of each factor. Instead, a series of regression models were carried out to determine the specific influence of each factor.

Table 4 shows the results of the regression models for the entire territory and for each cluster. The community of Madrid model (“C.Madrid”) reveals some interesting findings, as all components are not only significant, but also positively correlated. This means that higher scores on the three variables, including socioeconomic status, increase the risk of contracting COVID-19. This finding contradicts the existing theory on socioeconomic status and COVID-19 transmission [46]. However, it is important to mention that we are analyzing variables at the municipal level (district, in the case of the Madrid municipality), so intra-municipal variability is not taken into account. Therefore, our results indicate that municipalities with greater socioeconomic status (i.e., a greater proportion of workers and a higher average income) are likely to have more cases of COVID-19, which does not exclude the possibility that, within these municipalities, there may be some economic segregation in the risk of infection. Another important finding lies in the comparison of effect sizes. By working with standardized factors, it is possible to make comparisons between the regression parameters. Particulate matter and temperature scores had the greatest impact on the number of cases compared to the other two variables, with socioeconomic status having the least influence.

The partial models do not provide much information, given the parameters’ lack of statistical significance. The pollution and population density variables in the Madrid-Surroundings model and the control variable are the only significant coefficients. However, the results are not reliable, as the residuals are not normally distributed. This lack of significance can be attributed to two causes. First, there are only a small number of observations in each cluster. Second, the behavior of the target variable among the observations of each cluster is quite homogeneous. This last finding is consistent with Figure 3 and Scheffé’s test, which showed statistically significant differences in the number of cases among each cluster. 

The population control variable has the expected effect. It was included in the analyses to control for the effect of using an absolute rather than a relative measure. The regression models confirm that the more individuals in the territory, the more cases. 

## 5. Conclusions

The worldwide COVID-19 pandemic has led the scientific community to turn its attention to studying this phenomenon in the hopes of discovering which risk factors increase the population’s vulnerability. Environmental factors have been widely studied in this line of research. Despite the large number of studies conducted to date, the subject is far from fully explored, and still requires further examination. The present study aimed to create an exhaustive case study of the risk factors involved in the spread of COVID-19 in the community of Madrid during the first wave of the pandemic.

The study began with a principal component analysis to reduce a pool of thirteen identified risk factors down to three new components that explained 71% of the original variance. The new variables were conceptualized as pollutants and population density, particulate matter and temperature, and socioeconomic status. The newly derived variables were used to map the different ecosystems of the territory based on these risk dimensions. An ANOVA and a Scheffé´s multiple comparisons test confirmed the existence of significant differences in COVID-19 infection between the three clusters found. 

The analyses concluded with four regression models being used to determine the influence of each risk factor (one regression model for the whole territory, and one for each cluster). In the territory-wide model, all parameters were significant and showed a positive association with the number of infections. One of the most notable findings was that the pollution and climate components had a greater influence than the socioeconomic component on the overall model. In contrast to the above, the cluster models did not provide much information due to the non-significance of the parameters and the residuals lacking a normal distribution. Nevertheless, the models have a good fit according to the adjusted r-squared, AIC, BIC, and log likelihood. Therefore, the lack of significance may be due to the low sampling power of the models.

In considering the results, we draw two conclusions. First, pollutants and climate factors play an important role in increasing vulnerability to COVID-19 infection. Second, micro-level research is an important avenue of study. The first conclusion highlights the importance of caring for the environment, as it appears to be a more significant factor in COVID-19 transmission than even the socioeconomic status of a municipality. The second conclusion arises from the lack of explanatory power of partial models. Thus, researchers need to produce harmonized statistics at a higher level of granularity. This local knowledge is the key to designing strategies adapted to each territory and is consistent with the reality of their situations. At the same time, we must continue with pandemic surveillance to determine the consistency of the results over time, and thus improve our understanding of risk factors that impact the pandemic. To this end, we intend to replicate the present study in subsequent waves of the pandemic.

## Figures and Tables

**Figure 1 ijerph-18-09227-f001:**
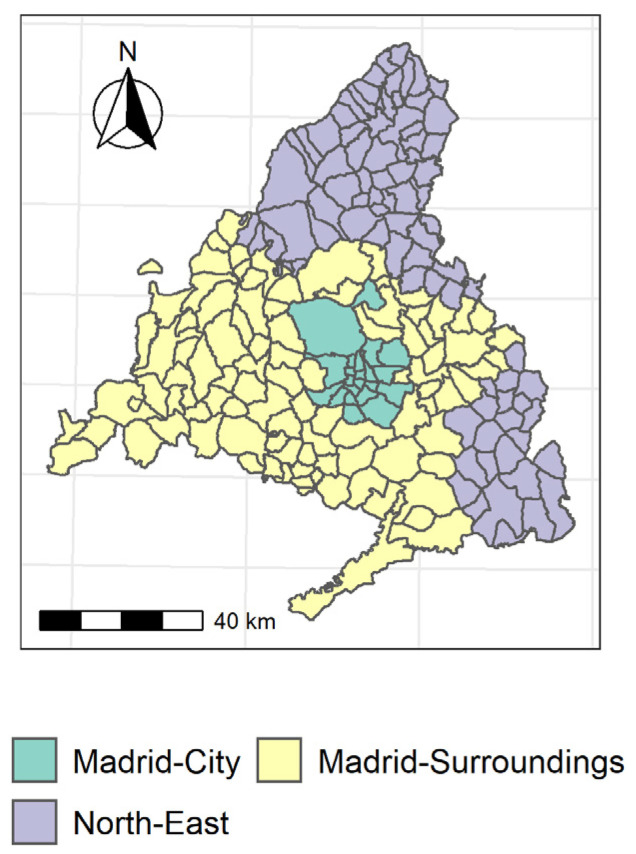
Cluster with components.

**Figure 2 ijerph-18-09227-f002:**
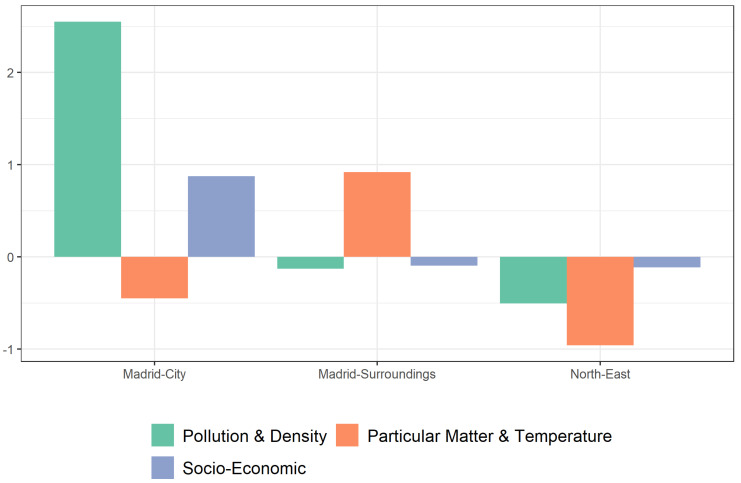
Components by clusters.

**Figure 3 ijerph-18-09227-f003:**
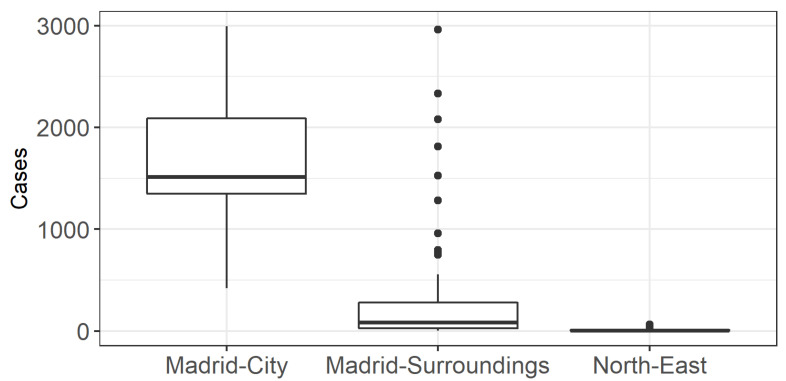
Boxplot of COVID-19 cases by cluster. The diagram represents the median value (line inside the box), the Q1 (lower boundary of the box), the Q3 (upper boundary of the box), the maximum and minimum (vertical line) and the outliers (the dots).

**Table 1 ijerph-18-09227-t001:** Summary of independent variables.

Variable	Measure	Data Source
**Social Dimension**		
Density	Population density	[10,11]
Age	Percentage of the population over 65 years old	[10,11]
Income	Per capita income by municipality from Estimate of the Municipal Gross Domestic Product	[12,13]
Workers	Percentage of workers supported by social security	[14,15]
**Pollution dimension**		
CO	Carbon monoxide, micrograms per cubic meter	[16,17]
NO	Nitrogen monoxide, micrograms per cubic meter	[16,17]
NO_2_	Nitrogen dioxide, micrograms per cubic meter	[16,17]
SO_2_	Sulphur dioxide, micrograms per cubic meter	[16,17]
Ozone	Ozone, micrograms per cubic meter	[16,17]
PM2.5	Particulate matter < PM2.5	[16,17]
Res.Sys.death	Respiratory system related death rate	[18]
**Climatological dimension**		
Temperature	Temperature April average level	[19]
Humidity	Humidity April average level	[19]
**Control variable**		
Population	Number of people	[11,20]

**Table 2 ijerph-18-09227-t002:** Component loadings.

	Pollution and Density	Particulate Matter and Temperature	Socioeconomic
SO_2_	**0.87**		
CO	**0.86**		
NO	**0.74**		
Density	**0.73**		0.42
PM2.5		**0.92**	
Ozone	−0.43	**−0.83**	
Temperature		**0.76**	
NO_2_	0.53	**0.68**	
Income			**0.87**
Workers			**0.76**
Humidity	−0.56	−0.50	
Age		−0.57	0.40
Resp.Sist.Deaths		0.54	

Note: Loads less than 0.4 have been suppressed. The most representative factor loadings are indicated in bold type.

**Table 3 ijerph-18-09227-t003:** Comparison of the mean values of the original variables by cluster.

Variable	Cluster			ANOVA (*p*-Value)
	Madrid-City	Madrid-Surroundings	North-East	
**Pollution Dimension**				
PM2.5	4.05	6.66	3.45	<0.01
*SD*	0.22	0.69	1.07	
SO_2_	4.33	1.29	1	<0.01
*SD*	1.56	0.50	0.00	
CO	50.40	0.87	0.45	<0.01
*SD*	12.33	0.69	0.19	
NO	4.15	1.82	1.00	<0.01
*SD*	4.57	0.90	0.01	
NO_2_	17.01	14.04	1.68	<0.01
*SD*	5.65	7.25	0.63	
Ozone	53.08	53.90	80.62	<0.01
*SD*	4.61	3.84	4.24	
**Social Dimension**				
Density	14.25	0.85	0.05	<0.01
*SD*	9.82	1.32	0.05	
Income	40,590.98	21,877.74	24,445.71	<0.01
*SD*	17,966.81	11,808.98	14,818.63	
%Workers	0.60	0.31	0.22	<0.01
*SD*	0.39	0.20	0.11	
**Climate Dimension**				
Humidity	65.96	71.16	75.98	<0.01
*SD*	0.00	3.56	0.54	
Temperature	9.69	12.83	10.89	<0.01
*SD*	0.00	0.92	1.60	

Dark gray, light gray and white are used to indicate that there are statistical differences between means. The darkest shade refers to the highest mean value, the light gray to the mean value and the white to the lowest mean value. When there are only statistically significant differences between two groups, only dark gray and light gray are used to distinguish the groups. Differences in means were tested by Scheffé´s methods at a significance level of 0.05.

**Table 4 ijerph-18-09227-t004:** Regression models with components.

	C.Madrid			North-East			Madrid-Surroundings			Madrid-City		
	Coef. (*SD*)	t	*p*-Value	Coef. (*SD*)	t	*p*-Value	Coef. (*SD*)	t	*p*-Value	Coef. (*SD*)	t	*p*-Value
(Intercept)	3.17 *** (0.08)	37.81	<0.01	1.17 ** (0.04)	3.13	<0.01	3.69 *** (0.42)	8.83	<0.01	6.11 *** (0.21)	28.79	<0.01
Pollution and Density	0.48 *** (0.08)	5.98	<0.01	−0.62 (0.49)	−1.26	0.21	0.77 * (0.32)	2.40	0.02	0.00 (0.14)	0.14	0.89
Particulate Matter and Temperature	0.75 *** (0.06)	12.35	<0.01	−0.10 (0.12)	−0.87	0.38	0.07 (0.44)	0.15	0.87	−0.13 (0.27)	−0.48	0.63
Socioeconomic	0.16 * (0.07)	2.31	0.02	0.15 (0.12)	−1.22	0.22	0.25 (0.21)	1.16	0.25	0.03 (0.76)	0.75	0.45
Population	0.00 *** (1.68)	11.66	<0.01	0.00 (0.00)	17.06	<0.01	0.00 *** (0.00)	7.78	<0.01	0.00 *** (7.58)	7.58	<0.01
Adjusted R-squared		0.82			0.82			0.66			0.81	
*p*-value		<0.01			<0.01			<0.01			<0.01	
White test (*p*-values)		<0.01			0.03			0.05			0.34	
Model Additional Information												
AIC		504.51			48.64			268.77			−2.93	
BIC		524.27			63.08			284.16			3.33	
Loglik		−246.25			−18.32			−128.38			7.46	

*** *p* < 0, ** *p* < 0.01, * *p* < 0.05.

## Data Availability

The open-source code and data required to replicate all analyses in this article will be available at a private GitHub repository (https://github.com/vicperez/R-script-and-data-of-Multivariate-analysis-of-risk-factors-during-the-first-wave-of-the-COVID-19-pa).

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
