# Peer review of "Multivariate Analysis of Risk Factors of the COVID-19 Pandemic in the Community of Madrid, Spain"

_ijerph, 2021, doi:10.3390/ijerph18179227_

Round 1

Reviewer 1 Report

I really like the paper but I am of the opinion that there are some issues that I believe need to be addressed before the paper can be published. Hence I am recommending major revision. Please, find below my detailed concerns.

- Abstract, line 21, I believe the word "incidence" should be "influence".

- Page 2, lines 72-73, it is affirmed that "a maximum likelihood principal component analysis (PCA, hereafter) with varimax rotation was applied". Even though Bartlett's test and KMO were calculated give adequate values, I wonder whether it is necessary to reduce the variables considering there are only 13 a small enough number to allow for the analysis to be done with the raw 13 variables.

- Page 2, lines 78-79 it is stated "To select the most appropriate number of clusters, the Elbow method was used". However there is no specification on how that was used so the choice of 3 groups for the municipalities seems arbitrary.

- Page 4, lines 164-165, the sentence "have recently discovered that the virus has a predilection for a certain blood group;" is missing something probably the citation.

- Page 5, line 187, the risk factors first dimension is said to be "social". However, in the following 189 line, the subsection heading says "Socio-economic". And in the following line 190 it says "social" again. I think the authors should be coherent and decide on one denomination.

- Page 5, lines 207-208, it is declared "Hawkins et al. [46] noted a strong correlation between socio-economic status and the number of cases." How does Hawkins et al. measure the socio-economic status?

- Also in page 5, lines 213-214, "Ahmed et al. [47] suggests that impoverished populations may have a poorer health status". How Ahmed et al. measure the "poverty"? Do the per capita income of the territory and the percentage of workers supported by social security cover those same factors?

- Page 8, Table 3. ANOVA p-values are included without any explanation or mention in the text. Also "Chromatic differences" are used to express "mean statistical significative differences". How are those differences expressed. More explanation is needed so the reader can understand the table.

- Page 9, Figure 4. A gradation of green in used. I would advise the authors to use a more friendly color and also another gradation so the differences can be easily distinguished.

- Page 10, regression analysis. I wonder why the authors have decided to select the raw number of cases as dependent variable, instead of considering a ratio by 100.000 inhabitants. Everybody is now familiar with that measure that would allow more significant comparisons. Also I am of the opinion that population density would work better than population.

- Page 11, line 361. "This result was repeated for two of the three regions of the territory." Those parameters are not significant, hence there is insufficient evidence to conclude that there is an effect at the population level.

- Page 12, Acknowledgments. Those acknowledgments are really funding for the paper, hence it should go in the corresponding place in page 11.

- Data and script should be provided as supplementary data for review.

- Several past participle are not correctly placed. For example, page 1, line 45, it is said: "factors considered, or the spatiotemporal frame studied" when it should be "considered factors, or the studied spatiotemporal frame".

Finally, there are many problem with references, both in the text and in the reference list. A careful review is needed. Here you have some examples:

- Page 3, lines 120-122, and page 4, line 123, author names should not be there as reference numbers are included. Also [28] has to be included alongside [27].

- Page 6, line 243, paper 13 and 6 are not related to covid.

- Reference 2, 23, 36, 52, have inverted commas in the paper title, while the rest of the references do not.

- Reference 3 has an "and" between the authors.

- Reference 26 is missing the link.

- Reference 28 is missing the journal.

- Is there an English version for reference 41? If that is the case I believe it would be better to include the English version over the Spanish one.

- Reference 44, there is a word in Spanish "Disponible en"

- Reference 50, the publication media (journal, book, web page...) does not appear.

Author Response

We want to thank you for your comments and suggestions concerning our paper,

We answer the points as follows:

1) Abstract, line 21, I believe the word "incidence" should be "influence".

 Done!

Abstract, line 21, the word incidence was replaced by influence.

2) Page 2, lines 72-73, it is affirmed that "a maximum likelihood principal component analysis (PCA, hereafter) with varimax rotation was applied". Even though Bartlett's test and KMO were calculated give adequate values, I wonder whether it is necessary to reduce the variables considering there are only 13 a small enough number to allow for the analysis to be done with the raw 13 variables.

The variable reduction technique (PCA) was applied for two reasons. The main reason is to reduce the number of variables in the models and thus improve the significance of the parameters by allowing a greater number of degrees of freedom. In addition, this technique made it possible to compare effects between complete dimensions and not between separate items. For consistency in the analyses, the cluster analysis was performed on the basis of the components, as the result was identical to that of including the variables.

3) Page 2, lines 78-79 it is stated "To select the most appropriate number of clusters, the Elbow method was used". However there is no specification on how that was used so the choice of 3 groups for the municipalities seems arbitrary.

Page 2, lines 78-80. More details on the selection criteria are included. In this regard we have carried out a graphical method (elbow method) to select the number of clusters.  According to the scree plot, three was considered the appropriate number of clusters in relation to the within-cluster sum of square decrease. If required, the figure can be included in the manuscript in the appendices section.

4) Page 4, lines 164-165, the sentence "have recently discovered that the virus has a predilection for a certain blood group;" is missing something probably the citation.

Pages 4-5, lines 174-175, the missing quotation was included.

5) Page 5, line 187, the risk factors first dimension is said to be "social". However, in the following 189 line, the subsection heading says "Socio-economic". And in the following line 190 it says "social" again. I think the authors should be coherent and decide on one denomination.

The text was harmonized by always using the term socioeconomic.

6) Page 5, lines 207-208, it is declared "Hawkins et al. [46] noted a strong correlation between socio-economic status and the number of cases." How does Hawkins et al. measure the socio-economic status?

Page 5, lines 218-222, the indicator used in the work of Hawkings et al. (46) has been included and some of the items it includes have been mentioned.

7) Also in page 5, lines 213-214, "Ahmed et al. [47] suggests that impoverished populations may have a poorer health status". How Ahmed et al. measure the "poverty"? Do the per capita income of the territory and the percentage of workers supported by social security cover those same factors?

Page 6, lines 228-230. The work of Ahmed et al. is theoretical in nature and does not operationalize the concept of poverty. It only limits itself to provide causal explanations of a possible COVID-19 infection gap according to socio-economic status.

Page 6, lines 232-238, As indicated in the paper, the income level is a good but not perfect approximation of the socioeconomic status of the neighborhood. Possible extreme values can skew the average value of the municipality. In addition, there are low-skilled, high-paying jobs that do not offer the same quality of life to workers as higher-skilled jobs. That is the reason why we have considered including the percentage of affiliated persons to make the socioeconomic characterization of the territory more robust.

8) Page 8, Table 3. ANOVA p-values are included without any explanation or mention in the text. Also "Chromatic differences" are used to express "mean statistical significative differences". How are those differences expressed. More explanation is needed so the reader can understand the table.

Page 9, lines 307-309. The table has been explained in more detail with reference to the p-value ANOVA and the post-hoc tests.

Page 10, lines 319-320. Footnote corrected and expanded to clarify the argument

9) Page 9, Figure 4. A gradation of green in used. I would advise the authors to use a more friendly color and also another gradation so the differences can be easily distinguished

Page 11, Figure 4. Figures 3 and 4 provided the same information. Figure 4 was redundant and was eliminated as it was less informative than Figure 3.

10) Page 10, regression analysis. I wonder why the authors have decided to select the raw number of cases as dependent variable, instead of considering a ratio by 100.000 inhabitants. Everybody is now familiar with that measure that would allow more significant comparisons. Also I am of the opinion that population density would work better than population.

Page 3, lines 106-113. The reason why is that on this occasion the relative ratio was less reliable than the absolute measure. According to the relative rate, the most affected populations were populations with a very small number of cases. As I say in the text: “This issue is highlighted by the fact that three of the six regions with the highest rates have fewer than seven cases to date (Valdaracete with six cases, and Berzosa del Lozoya and Braojos with less than five cases)”. To correct possible biases in the models by including an absolute and not a relative measure, the control variable was included: number of inhabitants of the territory.

11) Page 11, line 361. "This result was repeated for two of the three regions of the territory." Those parameters are not significant, hence there is insufficient evidence to conclude that there is an effect at the population level.

Page 12, line 360. Reviewing the models, it has been noticed that although there seems to be normality in the residuals of the model in question (Madrid-Surrondings) through the Q-Q plot, this was not verified by the tests. Consequently, it has been commented that the result is not valid.

12) Page 12, Acknowledgments. Those acknowledgments are really funding for the paper, hence it should go in the corresponding place in page 11.

Page 13, lines 413-414. The acknowledgement to the Comillas Pontifical University (PP2020_03) has been included under Funding.

13) Data and script should be provided as supplementary data for review.

The data and the R script are accessible in the Github repository: https://github.com/vicperez/R-script-and-data-of-Multivariate-analysis-of-risk-factors-during-the-first-wave-of-the-COVID-19-pa

14) Several past participles are not correctly placed. For example, page 1, line 45, it is said: "factors considered, or the spatiotemporal frame studied" when it should be "considered factors, or the studied spatiotemporal frame".

Done!

All the past participles of the work were corrected as indicated

15) Page 3, lines 120-122, and page 4, line 123, author names should not be there as reference numbers are included. Also [28] has to be included alongside [27].

Done!

Page 4, lines 129-133. The names of the authors have been deleted from the text.

16) Page 6, line 243, paper 13 and 6 are not related to covid.

It was a mistake to include it, the reference was deleted. Thanks so much

17) Reference 2, 23, 36, 52, have inverted commas in the paper title, while the rest of the references do not.

Done!

Reference 2, 23, 36, 52. The inverted commas were eliminated from these references

18) Reference 26 is missing the link.

Reference 26. The reference link has been included. Thanks so much

19) Reference 28 is missing the journal.

 Reference 28. The journal name has been included. Thank you

20) Is there an English version for reference 41? If that is the case I believe it would be better to include the English version over the Spanish one.

Reference 41. The reference in English has not been found. In addition, the website no longer exists. Consequently, it has been left as it is to respect the authorship of the idea.

21) Reference 44, there is a word in Spanish "Disponible en"

Reference 44. The Spanish words translated into English as: Available online

22) Reference 50, the publication media (journal, book, web page...) does not appear.

The publication medium of reference 50 was included.

This manuscript has been edited by the regular English editing service of MDPI.

Reviewer 2 Report

The manuscript is well structured and of adequate scientific level.

The bibliography is very up-to-date.

The subject is highly topical and is therefore a useful dissemination tool for the scientific community.

I suggest revising the English

Author Response

We want to thank you for your comments and suggestions concerning our paper. The manuscript has been edited by the regular English language editing service of MDPI.

Round 2

Reviewer 1 Report

I read the paper again with a great interest. I think the authors have improved their paper in such a way that it is almost ready for publication. I am recommending minor revision because I believe the readers would benefit from their including the scree plot in an appendix to support their deciding on 3 clusters.

Author Response

Thanks again for the comment. As recommended, the scree plot was included in the appendix and cited in the text.
